# Machine learning modeling of family wide enzyme-substrate specificity screens

**Samuel Goldman** [1,2], **Ria Das** [2,3], **Kevin K. Yang** [4], **Connor W. Coley** [2,3] *

**1** MIT Computational and Systems Biology, Cambridge, Massachusetts, United States of America, **2** MIT Chemical Engineering, Cambridge, Massachusetts, United States of America, **3** MIT Electrical Engineering and Computer Science, Cambridge, Massachusetts, United States of America, **4** Microsoft Research New England, Cambridge, Massachusetts, United States of America

* ccoley@mit.edu

## Abstract

Biocatalysis is a promising approach to sustainably synthesize pharmaceuticals, complex natural products, and commodity chemicals at scale. However, the adoption of biocatalysis is limited by our ability to select enzymes that will catalyze their natural chemical transformation on non-natural substrates. While machine learning and *in silico* directed evolution are well-posed for this predictive modeling challenge, efforts to date have primarily aimed to increase activity against a single known substrate, rather than to identify enzymes capable of acting on new substrates of interest. To address this need, we curate 6 different high-quality enzyme family screens from the literature that each measure multiple enzymes against multiple substrates. We compare machine learning-based compound-protein interaction (CPI) modeling approaches from the literature used for predicting drug-target interactions. Surprisingly, comparing these interaction-based models against collections of independent (single task) enzyme-only or substrate-only models reveals that current CPI approaches are incapable of learning interactions between compounds and proteins in the current family level data regime. We further validate this observation by demonstrating that our no-interaction baseline can outperform CPI-based models from the literature used to guide the discovery of kinase inhibitors. Given the high performance of non-interaction based models, we introduce a new structure-based strategy for pooling residue representations across a protein sequence. Altogether, this work motivates a principled path forward in order to build and evaluate meaningful predictive models for biocatalysis and other drug discovery applications.

## Author summary

Predicting interactions between compounds and proteins represents a long-standing dream of drug discovery and protein engineering. Robust models of enzyme-substrate scope would dramatically advance our ability to design synthetic routes involving enzymatic catalysis. However, the lack of standardization between compound-protein interaction studies makes it difficult to evaluate the generalizability of such models. In this work we take a critical step forward by standardizing high-quality datasets measuring enzyme-

**Data Availability Statement:** All code and relevant enzyme data can be found on GitHub at https://github.com/samgoldman97/enzyme-datasets. Code to reproduce models can be found at https://github.com/samgoldman97/enz-pred. Code to reproduce the reanlaysis on the kinase data from

Davis et al. can be found at https://github.com/samgoldman97/kinase-cpi-reanalysis.

**Funding:** SG was partially funded by the Machine Learning for Pharmaceutical Discovery and Synthesis consortium (mlpds.mit.edu). The funders had no role in study design, data collection and analysis, decision to publish, or preparation of the manuscript. All other authors received no specific funding for this work.

**Competing interests:** The authors have declared that no competing interests exist.

substrate interactions, outlining rigorous evaluations, and proposing a new way to integrate structural information into protein representations. In testing previous modeling approaches, we highlight a surprising inability of existing models to effectively leverage compound-protein interactions to improve generalization, which challenges a perception in the literature. This establishes future opportunities for model development and integration of enzyme-substrate scope models into computer-aided synthesis planning software.

This is a *PLOS Computational Biology* Methods paper.

## Introduction

Biology has evolved enzymes that are capable of impressively stereo-selective, regio-selective, and sustainable chemistry to produce compounds and perform reactions that are "the envy of chemists" [1–3]. Industrial integration of these enzymes in catalytic processes is transforming our bioeconomy, with engineered enzymes now producing various materials and medicines on the market today [2, 4]. As an exemplar of collective progress in biocatalysis and enzyme engineering, Huffman et al. impressively re-purposed the entire nucleoside salvage pathway for a high yield, 9-enzyme *in vitro* synthesis of the HIV nucleoside analogue drug, islatravir [5]. In an effort to make these types of pathways commonplace, there has been an explosion in new tools for automated computer-aided synthesis planning (CASP) that can include not only traditional organic chemistry reactions [6], but also enzymatic reactions, facilitating further growth of industrial biocatalysis [7–9].

Despite this progress in synthesis planning, suggesting an enzyme for each catalytic step in a proposed synthesis pathway remains difficult and limits the practical utility of synthesis planning software. Current enzyme selection methods often use simple similarity searches, comparing the desired reaction to precedent reactions in a database [10, 11]. Due to the often high selectivity of enzymes, proposed enzymes for a hypothetical reaction step often suffer from low catalytic efficiency. In the extreme case, the proposed enzyme may have zero catalytic effect on the substrate of interest, despite showing moderate activity on a similar natural substrate. Thus, the specificity of enzymatic catalysis can be a double edged sword [12]. As an example, the phosphorylation step in the islatravir synthesis of Huffman et al. required screening a multitude of natural kinase classes to find an enzyme capable of phosphorylating the desired substrate with sufficient activity for subsequent directed evolution [5]. In the less extreme case, the enzyme of interest may have moderate activity but suffer from low initial substrate loadings, proceed slowly, require higher catalyst loadings, and produce low yields [2]. Nevertheless, an enzyme with moderate activity can serve as a "hook" for further experimental optimization and directed evolution efforts.

Machine learning and predictive modeling provide an avenue to accelerate long development cycles and identify enzymes with both initial activity and high efficiency. Sequence-based machine learning methods have already been utilized in "machine learning guided directed evolution" (MLDE) campaigns to help guide the exploration of sequence space toward desirable protein sequences [13–15]. MLDE demonstrations often follow a similar paradigm: given a screen of a single enzyme with mutations at select positions, predict the function or activity of enzymes with new mutations. Further developments in pretrained machine

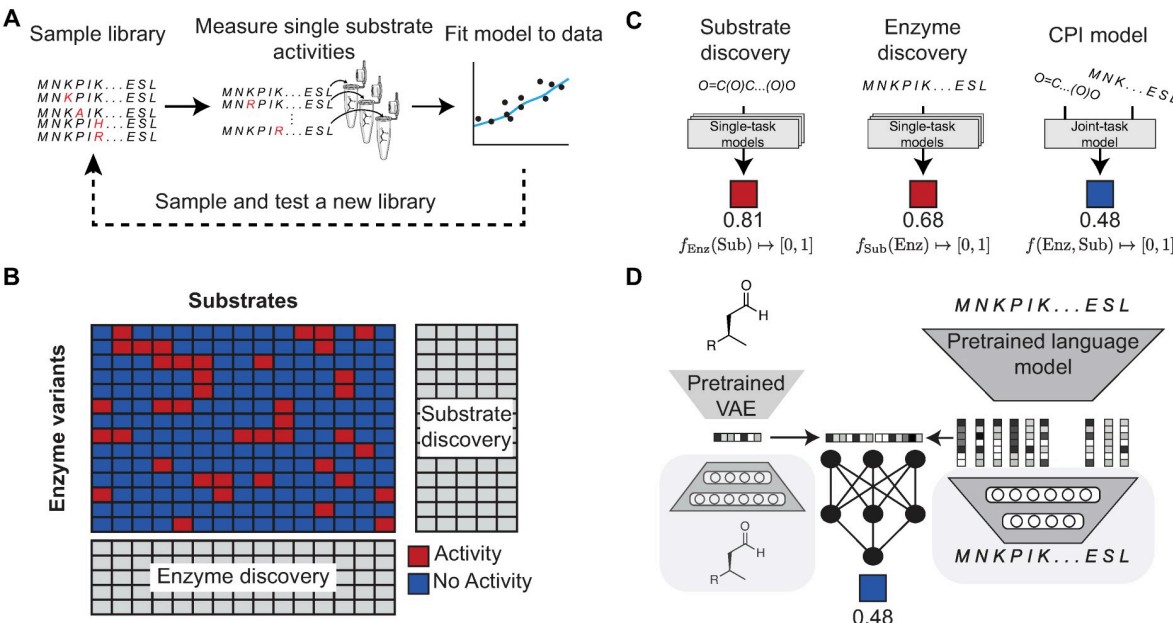

**Fig 1. Enzyme-substrate interaction modeling strategies.** **(A)** Current machine learning-directed evolution strategies, which involve design-build-test-model-learn cycles measuring protein variant activity on a single substrate of interest. **(B)** The "dense screen" setting where homologous enzyme variants from one protein family are profiled against multiple substrates. In this setting, we can aim to generalize to either new enzymes ("enzyme discovery") or new substrates ("substrate discovery"). **(C)** Three different styles of models evaluated in this study, where single task models independently build predictive models for rows and columns from panel (B), whereas a CPI model takes both substrates and enzymes as input. **(D)** An example CPI model architecture where pretrained neural networks extract features from the substrate and enzyme to be fed into a top-level feed forward model for activity prediction.

learning models can now provide meaningful embeddings at single amino acid positions that capture contextual and structural information about the protein from sequence alone [16–21]. Pre-trained machine learning models of protein sequences, specifically masked language models adopted from natural language processing [22] are trained to predict the identity of "masked" input tokens (i.e. amino acids). In doing so, the model learns a meaningful intermediate representation of the protein and distill important structural context around each amino acid position. This intermediate layer can then be extracted and treated as a fixed embedding of the protein [23]. These pre-trained embeddings have proved especially useful for the application of MLDE in low-N settings where the number of protein measurements is small (Fig 1A) [24, 25]. Altogether, these approaches provide a way to improve the efficiency of an enzyme given examples of other enzymes with activity on the substrate of interest. However, this paradigm does not extend to meet the the challenge of identifying a "hook" enzyme with sufficient initial activity on a non-native substrate, nor can current approaches incorporate information from enzymes measured on *other* substrates.

Instead, work to date to expand the substrate scope of an enzyme class of interest often relies upon time consuming and *ad hoc* rational engineering based upon structure [26, 27], simple similarity searches between the native substrate and desired substrate of interest [28], or trial-and-error experimental sampling [29]. Once an enzyme with some activity for a substrate of interest is found practitioners can resume directed evolution strategies similar to those described above to increase efficiency [29–32].

The last strategy of experimental sampling often involves broad metagenomic sampling [33–36], where homologous sequences are chosen for testing [35, 37, 38]. Researchers will test a diverse set of "mined" enzymes for activity against a panel of substrates containing the

relevant reactive group. This experimental screening of enzymes against substrates closely mirrors the data setting involved in discovering selective inhibitors in drug discovery, where a panel of similar proteins such as kinases [39] or deubiquinating proteins [40, 41] are screened against a family of compounds. While some work in this field of compound protein interactions (CPI) has attempted to model the drug discovery framing of this problem [42, 43], the CPI modeling framework has not yet been extended to enzyme promiscuity and there exist few curated datasets to probe our ability to learn from enzyme screens.

In this work, we model enzyme-substrate compatibility as a compound-protein interaction task using a carefully curated set of recent metagenomic enzyme family screens from the literature. We compare state of the art predictive modeling using pretrained embedding strategies (for both small molecules and proteins) and CPI prediction models. Surprisingly, we find that predictive models trained jointly on enzymes and substrates fail to outperform independent, single-task enzyme-only or substrate-only models, indicating that the joint models are incapable of learning interactions. To determine whether this is a quirk specific to our datasets, we reanalyze a recent CPI demonstration and find that this trend generalizes beyond enzyme-substrate data to CPI more broadly: learning interactions from protein family data to go beyond single-task models remains an open problem. Finally, we introduce a new pooling strategy specific to metagenomically-sampled enzymes using a multiple sequence alignment (MSA) and reference crystal structure to enhance enzyme embeddings and improve model performance on the task of enzyme activity prediction. Collectively, this work lays the foundation and establishes dataset standards for the construction of robust enzyme-substrate compatibility models that are needed for various downstream applications such as biosynthesis planning tools.

## Results

### Data summary

In order to systematically evaluate our ability to build models over enzyme-substrate interactions, we first need high quality data. Databases of metabolic reactions such as BRENDA [44] describe large numbers of known enzymatic reactions, but are collected from many sources at different concentrations, temperatures, and pH values. Instead, we turn to the literature to find high-throughput enzymatic activity screens with standardized procedures, i.e., exhibiting no variation in the experiments besides the identities of the small molecule and enzyme. We extract amino acid sequences and substrate SMILES strings from six separate studies measuring the activity of halogenase [45], glycosyltransferase [46], thiolase [47], ß-keto acid cleavage enzymes (BKACE) [48], esterase [49], and phosphatase enzymes [50] which cover between 1, 000 and 36, 000 enzyme-substrate pairs (Table 1). Enzymatic catalysis (e.g., yield, conversion,

**Table 1. Summary of curated datasets with the number of unique enzymes, unique substrates, and unique pairs in each dataset in addition to an exemplar structure for the protein family.**

| Dataset | # Enz. | # Sub. | Pairs | PDB Ref. |
|---|---|---|---|---|
| Halogenase [45] | 42 | 62 | 2,604 | 2AR8 |
| Glycosyltransferase [46] | 54 | 90 | 4,298[a] | 3HBF |
| Thiolase [47] | 73 | 15 | 1,095 | 4KU5 |
| BKACE [48] | 161 | 17 | 2,737 | 2Y7F |
| Phosphatase [50] | 218 | 165 | 35,970 | 3L8E |
| Esterase [49] | 146 | 96 | 14,016 | 5A6V |
| Kinase (inhibitors) [39] | 318 | 72 | 22,896 | 2CN5 |

[a] While most datasets we use test all combinations, Yang et al. do not report experiments for some enzyme by substrate interactions

activity) in each study was measured using some combination of coupled assay reporters, mass spectrometry, or fluorescent substrate readouts. Data was binarized such that every measured pair is either labeled as active (1) or inactive (0) at thresholds according to standards described in the original papers (Methods). We conceptualize these datasets as "dense screens" insofar as each dataset represents a number of enzyme and substrate pairs measured against each other resembling a grid (Fig 1B). While several of the experimental papers presenting these datasets include their own predictive modeling [46–48], these demonstrations are not systematically compared with standard splits and are not easily evaluated against new methods due to varied data formats. In compiling these, we expose new datasets to the protein machine learning community. Additional details can be found in the Methods.

In order to evaluate the ability of data-driven models to generalize beyond the set of screened enzymes and substrates, we examine extrapolation in two directions: enzyme discovery or substrate discovery (Fig 1C). In the former, we consider the setting of a practitioner who is interested in finding enzymes with activity on some set of substrates they have already measured. This parallels the setting where machine learning directed evolution may also be applied, such as increasing the efficiency of an enzyme (Fig 1A). On the other hand, for substrate discovery, we are interested in predicting which enzymes from an already-sampled set will act on a new substrate that has not already been measured, a formulation specifically relevant to synthesis planning. We omit the more difficult problem of generalizing to new substrates and new enzymes simultaneously; we posit that we must first be able to generalize in each direction separately in order to generalize jointly and note that empirical performance on joint generalization in compound protein interaction is lower in previous CPI studies [42]. We do not consider the task of interpolation within a dense screen (Fig 1B), as this does not reflect any realistic experimental application.

## Models

We aim to build a modeling pipeline that accepts both an enzyme and substrate and predicts sequence, with enzymes and substrates specified by sequence and SMILES strings respectively. To featurize enzymes, we turn to pre-trained protein language models, specifically the currently state-of-the-art ESM-1b model [20]. Pre-trained representations are well-suited to low data tasks and have been applied to protein property prediction [20, 21, 24] as well as compound protein interaction [51]. While there has been an explosion in available pre-trained protein representations including UniRep [21], SeqVec [52], MT-LSTM [16, 17], ESM-1b stands out in its performance on contact prediction tasks, ease of use, and also ability to effectively predict the functional effect of sequence variations, likely enabled by its comparatively large scale (i.e., number of parameters and training sequences) [20]. To featurize substrates, we test two primary featurizations: a pretrained Junction-Tree Variational Auto-Encoder (JT-VAE) [53] and the widely used Morgan circular fingerprints (1024 bits) [54]. Despite pre-trainining compound representations having only marginal benefits on property prediction tasks [55], a recent CPI study [42] extracted compound representations from a pre-trained JT-VAE model and utilized these to identify new kinase inhibitors. Due to the closeness in our proposed task, we follow their methodology and extract the same embeddings for substrates to compare against Morgan fingerprints.

If a single model is able to successfully learn interactions and leverage the full "dense" dataset, it should be able to take as input both enzyme and substrate representations and outperform smaller, single-task models that either use only enzyme inputs or use only substrate inputs (Fig 1C). To attempt to model interactions, we consider two simple top models inspired by the CPI literature [42, 43, 51] that either (1) concatenate the representations of the enzyme

and substrate before applying a shallow feed forward neural network or (2) project the representations of the enzyme and substrate to smaller and equal length vectors using a shallow multi-layer perceptron (MLP) before taking their dot product (Fig 1D). To evaluate these CPI based architectures, we consider three other model classes for comparison: baselines utilizing simple similarity across enzymes and substrates for prediction; multi-task models that learn to predict activity for enzymes against substrates simultaneously but without any feature information about the substrates themselves (and *vice versa* for substrate discovery); and single-task models with no information sharing across substrate (enzyme) tasks. We refer the reader to S1 Text for a more complete description of all model classes evaluated (Table C in S1 Text).

## Enzyme discovery

We test these various featurizations and model architectures first on the task of enzyme discovery. To do so, we hold out a fraction of the enzymes as a test set and use the training set to make predictions about the interactions between the held out enzymes and the known substrates in the data set. For each dataset, we train the CPI model architectures described above jointly on the entire training set. To test whether CPI models are able to learn interactions, we also train several smaller "single-task" models. These single-task models are specific to each substrate and accept only an enzyme sequence as input. If models are in fact able to learn interactions, the CPI models should outperform the single-task models given their access to more substrate measurements for each enzyme (Figs 2A and 1C).

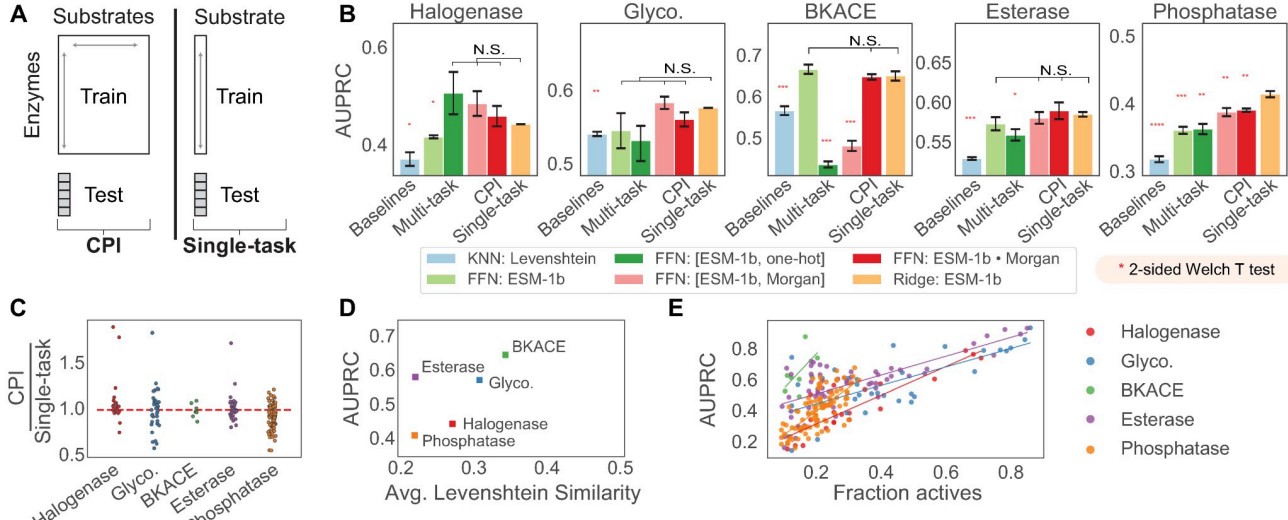

**Fig 2. Assessing enzyme discovery in family wide screens. (A)** CPI models are compared against the single task setting by holding out enzymes for a given substrate and allowing models to train on either the full expanded data (CPI) or only data specific to that substrate (single-task). **(B)** AUPRC is compared on five different datasets, arranged from left to right in order of increasing number of enzymes in the dataset. Baseline models are compared against multi-task models, CPI models, and single-task models. K-nearest neighbor (KNN) baselines are calculated using Levenshtein edit distances to compare sequences; multi-task models use a shared feed forward network (FFN) to compute predictions against all substrate targets, CPI models utilize FFN with either concatenation ("[{prot repr.}, {sub repr.}]") or dot product interactions ("{prot repr.}•{sub repr.}"), and ridge regression is used for single-task models. ESM-1b features indicate protein features extracted from a masked language model trained on UniRef50 [20]. Halogenase and glycosyltransferase datasets are evaluated using leave-one-out splits, whereas BKACE, phosphatase, and esterase datasets are evaluated with 5 repeats of 10 different cross validation splits. Standard error bars indicate the standard error of the mean of results computed with 3 random seeds. Each method is compared to the single-task L2-regularized logistic regression model ("Ridge: ESM-1b") using a 2-sided Welch T test, with each additional asterisk representing significance at [0.05, 0.01, 0.001, 0.0001] thresholds respectively after application of a Benjamini-Hochberg correction. **(C)** Average AUPRC on each individual "substrate task" is compared between compound protein interaction models and single-task models. Points below 1 indicate substrates on which single-task models better predict enzyme activity than CPI models. CPI models used are FFN: [ESM-1b, Morgan] and single-task models are Ridge: ESM-1b. **(D)** AUPRC values from the ridge regression model are plotted against the average enzyme similarity in a dataset, with higher enzyme similarity revealing better predictive performance. **(E)** AUPRC values from the ridge regression model broken out by each task are plotted against the fraction of active enzymes in the dataset. Best fit lines are drawn through each dataset to serve as a visual guide.

To evaluate each dataset, we calculate the area under the precision recall curve (AUPRC), computed separately for each substrate column and subsequently averaged. AUPRC is able to better differentiate model performance on highly imbalanced data than the area under the receiver operating curve (AUROC), which overvalues the prediction of true negatives. Further, AUPRC does not require choosing a threshold to call hits like other metrics like the Matthews Correlation Coefficient. We optimize model hyperparameters on the thiolase dataset [47] prior to training and evaluating on the remaining five datasets. We additionally report benchmarking performance for the thiolase dataset (Tables E and F in S1 Text).

We observe that our supervised models using pretrained protein representations are in fact able to outperform a nearest neighbor sequence-similarity baseline ("KNN: Levenshtein") that uses the Levenshtein distance, a simple unweighted global alignment distance used in recent protein engineering studies [24, 56], to predict held out enzyme activity (Fig 2B and Tables E and F in S1 Text). This affirms the potential of representation learning to improve prediction and protein engineering tasks.

Surprisingly, however, CPI models do not outperform single-task models trained with simple logistic regression ("Ridge: ESM-1b") (Fig 2B). Multi-task models offer a slight benefit on the halogenase dataset, but fail to outperform single-task models across the other four enzyme families tested. Further, upon closer inspection, the comparative performance of CPI based models on the halogenase dataset seem to be driven by relative performance increases only on a small number of substrate tasks as demonstrated by the upper outliers in Fig 2C. This is despite the CPI (and multi-task) models having access to a larger number of enzyme-substrate interactions for training. In fact, models trained with CPI can at times perform worse than models that predict the activity of enzymes on each substrate task independently (Fig 2C), indicating an inability to learn interactions from the dense screens collected.

The enzymes within each dataset were sampled with different levels of diversity by the studies' original authors. The phosphatase dataset represents a diverse super-family of enzymes [50], whereas the BKACE dataset represents a more narrowly sampled domain of unknown function (unknown prior to the experimental screen) [48]. We find that the average pairwise Levenshtein similarity between sequences in the dataset is in fact correlated with performance differences across datasets, such that more similar datasets seem to be easier to predict (Fig 2D).

In addition to intra-dataset diversity, we also hypothesized that the balance, or fraction of active enzymes, observed for each enzyme could partially explain the observed performance. Plotting the AUPRC metric as a function of number of active enzyme-substrate pairs reveals a strong positive correlation, validating that the number of hits observed in the training set will largely determine the success of the model in generalizing beyond the training set (Fig 2E). This is equally a function of both the models and the AUPRC metric, which follows a similar trend for random guesses that favor the majority binary class.

## Substrate discovery

We next evaluate generalization in the direction of held out substrates, repeating the same procedures as above. In this case, we restrict our analysis only to the glycosyltransferase and phosphatase datasets where the number of substrates is > 50, using the halogenase dataset to tune hyperparameters for each model. We report the results comparison on the halogenase dataset (Tables H and G in S1 Text).

Similar to our conclusions in the case of enzyme discovery, we find that the CPI architectures are not able to outperform simpler, single-task logistic regression models with Morgan fingerprints (Fig 3 and Fig E in S1 Text and Tables G and H) in S1 Text.

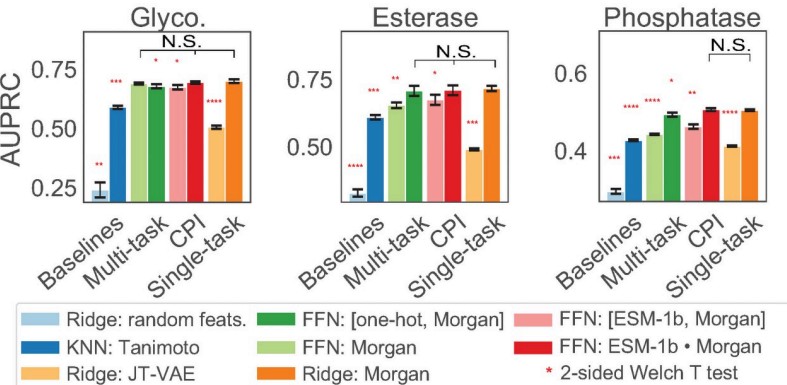

**Fig 3. Assessing substrate discovery in family wide screens.** CPI models and single task models are compared on the glycosyltransferase, esterase, and phosphatase datasets, all with 5 trials of 10-fold cross validation. Error bars represent the standard error of the mean across 3 random seeds. Each model and featurization is compared to "Ridge: Morgan" using a 2-sided Welch T test, with each additional asterisk representing significance at [0.05, 0.01, 0.001, 0.0001] thresholds respectively, after applying a Benjamini-Hochberg correction. Pretrained substrate featurizations used in "Ridge: JT-VAE" are features extracted from a junction-tree variational auto-encoder (JT-VAE) [53]. Two compound protein interaction architectures are tested, both concatenation and dot-product, indicated with "[{prot repr.}, {sub repr.}]" and "{prot repr.}•{sub repr.}" respectively. In the interaction based architectures, ESM-1b indicates the use of a masked language model trained on UniRef50 as a protein representation [20]. Models are hyperparameter optimized on a held out halogenase dataset. AUCROC results can be found in Fig D in S1 Text.

Curiously, in both the enzyme discovery (Figs I, J, H, G, and K) in S1 Text and substrate discovery (Figs M, L, and N in S1 Text) settings, predictions made by CPI models exhibit far more "blocky" characteristics than the respective single task models: when extrapolating to new enzymes, the prediction variance is not sensitive to the paired substrate for CPI models. This indicates that our CPI models struggle to condition their predictions to new enzymes (substrates) based upon the substrate (enzyme) pairing.

## Reanalysis of kinase inhibitor discovery

The results for enzyme-substrate activity prediction demonstrate that models designed to learn interactions are seemingly unable to do so in a manner that improves generalization. We therefore wondered to what extent this failing was specific to enzyme-substrate data, as opposed to being symptomatic of a broader problem and shortcoming in the CPI field, including drug discovery. To interrogate this, we re-analyze models from a recent study leveraging an inhibitor screen against the human kinome to discover new inhibitors against tuberculosis [42]. In their study, Hie et al. train CPI models on a dense screen of 442 kinases against 72 inhibitors [39] using concatenated pretrained protein and pretrained substrate features as the input to multi layer perceptrons (MLP), Gaussian processes (GPs), or a combination of the two (GP + MLP), the combination being their most successful (Methods). Unlike the binary classification enzyme activity setting, they predict continuous $K_d$ values.

We compare the MLP and GP+MLP models using pretrained representations against a number of single-task models on two settings matching the original study: drug repurposing and drug discovery. Drug repurposing is analogous to enzyme discovery where certain proteins are held out; drug discovery is analogous to substrate discovery where certain compounds are held out. Single-task models are not presented with training data on other kinase-compound pairs and are therefore unable to learn interactions in a generalizable manner. In addition to the single-task MLP and GP+MLP models, we evaluate a simple single-task, L2-regularized linear regression model ("Ridge") using Morgan fingerprint features rather than

JT-VAE features. In concordance with our results on enzymatic data, we find that single-task models consistently outperform CPI based models in terms of Spearman correlation coefficient between true and predicted $K_d$ on both repurposing (Fig 4A) and discovery tasks (Fig 4B). This shows that ablating interactions by training single-task models can increase performance over GP+MLP models.

Still, increased rank correlation between predictions and true $K_d$ values does not necessarily equate to the ability to select new inhibitors or new drugs. To directly test this, we repeat the retrospective kinase-inhibitor lead prioritization experiments conducted in the original analysis. Models are trained on a set of kinase-inhibitor pairings and used to rank new kinase-inhibitor pairings. The acquisition preference is informed by predictions and, if applicable, predicted uncertainty (Methods). When acquiring either 5 or 25 new data points in cross validation, a single-task ridge regression model with equivalent pretrained features is able to outperform both CPI based models (Fig 4C and 4D). Our findings are retrospective in nature and do not negate the value of prospective experimental validation [42], but rather make clear that the field requires new methods to leverage the rich information contained in protein, small molecule interaction screens and truly learn interactions. This further reinforces the necessity for simple baseline models in protein engineering studies [57, 58].

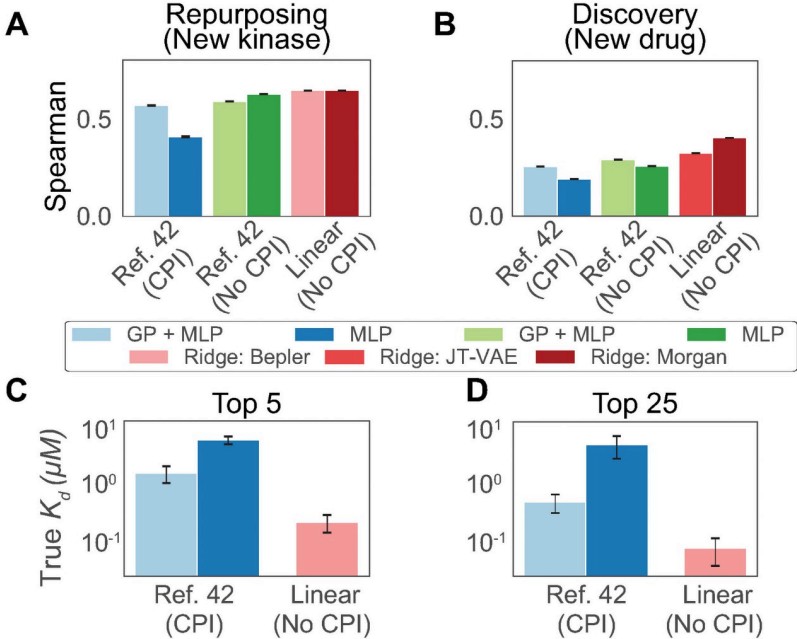

**Fig 4. Evaluating single-task models on kinase repurposing and discovery tasks.** Kinase data from Davis et al. is extracted, featurized, and split as prepared in Hie et al. Multilayer perceptrons (MLP) and Gaussian process + multilayer perceptron (GP+MLP) models are employed. We add variants of these models without CPI training separate single-task models for each enzyme and substrate in the training set, as well as linear models using both pretrained featurizations ("Ridge: JT-VAE") and fingerprint based featurizations of small molecules ("Ridge: Morgan"). Spearman correlation is shown for **(A)** held out kinases not in the training set and **(B)** held out small molecules not in the training set across 5 random initializations. **(C)** We repeat the retrospective evaluation of lead prioritization. The top 5 average acquired $K_d$ values are shown for the CPI models in Hie et al. compared against a linear, single-task ridge regression model using the same features. **(D)** The top 25 average acquired $K_d$ values are shown.

## Improving enzyme discovery models

Given that single-task models appear to match or even outperform models design for CPI, we next asked if we could improve their generalization in the enzyme discovery direction by leveraging the relationship between different protein sequences within the dataset. That is, working within a single family of proteins *should* be more conducive to generalizations. To directly impart this structural bias on our models, we considered how the construction of the pretrained representation for each protein could be modified. Pretrained language models produce a fixed dimensional embedding at each amino acid position in the protein. To collapse this into a fixed-length protein-level embedding, the *de facto* standard is to compute the mean embedding across the length of the sequence [16, 17, 20, 21, 24]. However, for locally-defined properties, such as enzymatic catalysis or ligand binding at an active site, this mean pooling strategy may be sub-optimal [59]. Previous approaches have largely considered deep mutational scans with few mutations at carefully selected positions [14]. In these settings, mean pooling strategies may be a good approximation of local protein structural changes, as embeddings at distal positions from the mutation would be nearly constant across protein variants. In our setting, however, we have metagenomically sampled sequences with large insertions and deletions, which presents an ideal testing ground to evaluate pooling strategies.

We test 3 alternative pooling strategies to mean pooling, where we first compute a multiple sequence alignment (MSA) and pool only a subset of residues in each sequence corresponding to a subset of columns in the MSA (Fig 5A). We rank order the columns in the MSA to be pooled based upon the (i) proximity to the active site of a single "reference structure" (Section "Active site pooling") (ii) coverage (i.e., pooling columns with the fewest gaps), or (iii) conservation (i.e., pooling columns that have the highest frequency of any single amino acid type)

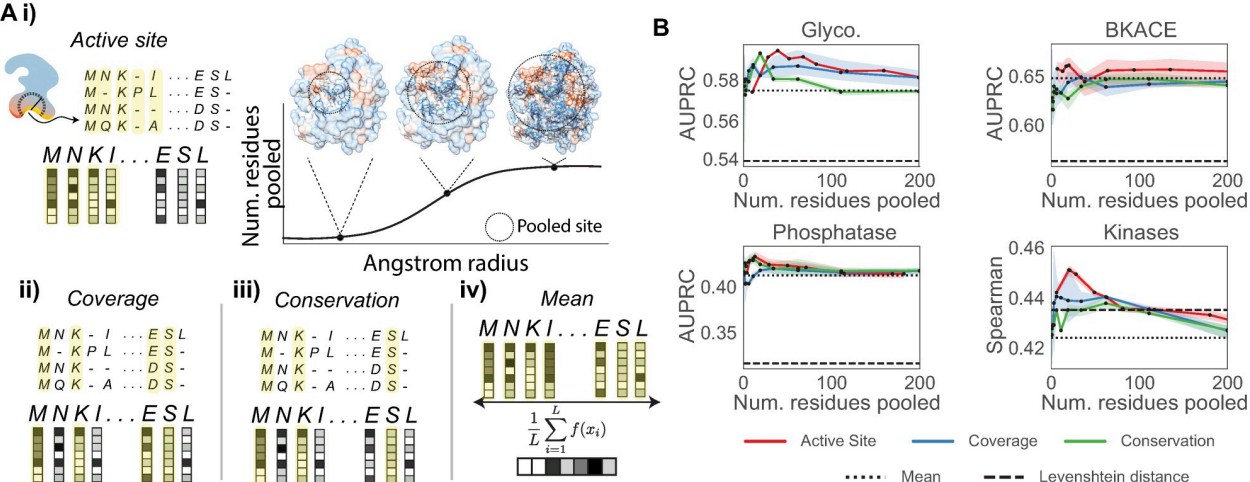

**Fig 5. Structure-based pooling improves enzyme activity predictions. (A)** Different pooling strategies can be used to combine amino acid representations from a pretrained protein language model. Yellow coloring in the schematic indicates residues that will be averaged to derive a representation of the protein of interest. **(i)** We introduce active site pooling, where only embeddings corresponding to residues within a set radius of the protein active site are averaged. By increasing the angstrom radius from the active site, we increase the number of residues pooled. Crystal structures shown are taken from the BKACE reference structure, PDB: 2Y7F rendered with Chimera [60]. **(ii, iii)** We also introduce two other alignment based pooling strategies: coverage and conservation pooling average only the top-*k* alignment columns with the fewest gaps and highest number of conserved residues respectively. **(iv)** Current protein embeddings often take a mean pooling strategy to indiscriminately average over all sequence positions. **(B)** Enzyme discovery AUPRC values are computed for various different pooling strategies. Each strategy is tested for different thresholds of residues to pool, comparing against both KNN Levenshtein distance baselines and a mean pooling baseline. The same hyperparameters are used as set in Fig 2 for ridge regression models. The kinase repurposing regression task from Hie et al. is shown with Spearman's $\rho$ instead of AUPRC as interactions are continuous, not binarized. All experiments and are repeated for 3 random seeds.

(Table 1). To expand this analysis beyond catalysis to drug discovery, we also consider a portion of the kinase inhibitor dataset from Davis et al. [39], subsetted down to a single kinase family (PF00069), rather than the whole human kinome (Table 1). We compare these pooling strategies across the enzyme discovery datasets tested, as well as the protein-inhibitor kinase dataset. We use ridge regression models with pretrained ESM-1b [20] embeddings, and split the data as in the enzyme discovery setting, varying only the pooling strategy from our previous analysis (Fig 2). In the case of the kinase regression dataset, we use the Spearman rank correlation to evaluate performance.

In all cases tested, the active site pooling performance peaks when pooling only a small number of residues around the active site ($< 60$ amino acids), showing gains in performance over other pooling strategies as well as the mean pooling baseline (Fig 5B). This corresponds to a distance of $< 10$ angstroms away from the active site (Fig 5Ai). This may indicate an optimal range at which residue positioning is relevant to promiscuity. In the case of the kinases, the Levenshtein distance baseline outperforms the mean pooling method with respect to Spearman rank correlation, but using active site aware pooling outperforms both. We find that the performance increase is steeper in the kinase repurposing setting compared to other datasets, potentially due to both the regression nature of the dataset and also the non-dynamic binding of compound inhibitors in comparison to the other tasks, which focus on enzymatic catalysis not protein inhibition. Interestingly, for the halogenase dataset, no alternative pooling strategies outperform mean pooling (Fig F in S1 Text), likely because the halogenase enzymes have high variance in solubility, a global property that could be driving enzyme activity [45]. Similarly, for the esterase dataset, coverage pooling is far more effective, indicating that a combination of targeted pooling residue strategies may be most effective (Fig F in S1 Text). While performance gains from active site aware pooling are modest, this strategy provides a simple but principled way to incorporate a structural prior into enzyme prediction models, particularly for metagenomic data with high numbers of sequence indels which may introduce unwanted variance into a mean pooled protein representation.

## Discussion

Data-driven models of enzyme-substrate compatibility have the potential to drive new insights in basic biology research and also to accelerate engineering efforts focused on the design of new enzymatic synthesis routes. In addition, the same classes of models can be used for compound-protein interaction prediction for both drug discovery and drug repurposing efforts.

In this work, we take a critical step toward opening up this suite of problems to machine learning researchers by providing several high quality, curated datasets and standardized splits to evaluate model performance and generalizability. While the small number of unique enzymes and unique substrates in each dataset makes quantitative performance sensitive to hyperparameters and dataset splitting decisions, this collection of data is an essential starting point to develop new modeling strategies and motivate future, higher throughput enzyme activity screening.

Our experiments show that pretrained representations for proteins, coupled with structure-informed pooling techniques, can go beyond standard sequence similarity based approaches to predict protein function, an exciting demonstration of how representation machine learning can impact protein engineering. Nevertheless, despite this excitement, our analysis makes clear that current CPI modeling strategies cannot consistently leverage information from multiple substrate measurements effectively, a problem broadly applicable to CPI models. That is, models designed to learn interactions do not outperform single-task models.

## Conclusion

To predict enzyme-substrate compatibility or design selective inhibitors against a protein family, we need new strategies to jointly embed proteins and compounds to enable more robust extrapolation to new combinations thereof. Such a scheme would allow learned interactions to be more explicitly transferred from larger databases onto smaller, but higher quality screen. This will be an exciting frontier in protein and compound representation learning, as the field seeks to go beyond protein structural prediction to protein function prediction. Further, with the exception of our structure informed (MSA-informed) pooling, our analysis remains sequence based. The relative performance of structure-based tools such as molecular dynamics for the prediction of enzyme-substrate scope remains an exciting question that this data, coupled with recent advances in protein structure prediction [61–63], can help to address.

## Methods

### Dataset preparation

Each dataset is collected from their respective papers [39, 45–50]. Activity binarizations are chosen to closely mirror the original dataset preparation with exact cutoff thresholds described in S1 Text. Additionally, certain enzymes were filtered based upon low solubility or activity that may result from screening decisions (see S1 Text).

### Davis kinase filtering

Kinases used in reanalysis of Hie et al. are tested exactly as prepared [42]. To evaluate structure based pooling using this dataset, we further subset the original dataset such that each entry only contains one domain from the PFAM family, PF00069, described in the SI with dataset statistics in Table 1.

### Hyperparameter optimization

All hyperparameters are set on a held out enzyme-substrate dataset using the hyperparameter optimization framework Optuna [64]. Hyperparameter optimization is set using up to 10 trials of leave-one-out cross validation on the thiolase dataset and halogenase dataset for enzyme discovery and substrate discovery tasks, respectively. Hyperparameters are chosen to maximize the average area under the precision recall curve. For nearest neighbor models, the number of neighbors is treated as a hyperparameter between 1 and 10.

For logistic ridge regression models, the regularization coefficient, $\alpha$ is set from $\{1e - 3,$ $1e - 2, 1e - 1, 1e0, 1e1, 1e2, 1e3, 1e4\}$. For both feed-forward dot product and concatenation models, hyperparameters for dropout ($[0, 0.2]$), weight decay ($[0, 0.01]$), hidden dimension ($[10, 90]$), layers ($[1, 2]$), and learning rate ($[1e - 5, 1e - 3]$) are chosen. All neural network models are trained for 100 epochs using the Adam optimizer and Pytorch [65].

For linear ridge regression used in reanalysis of kinase data, a default hyperparameter regularizer value of $\alpha = 1e1$ is set.

### Evaluation metrics

To evaluate models in the enzyme discovery direction, activity on each substrate is considered to be its own "task". The data is divided up into a set number of folds, and models are retrained to make predictions on each held out fold. A single, separate AUPRC value is computed for the activity on each substrate task and then averaged across substrate tasks. AUPRC values are computed using the average precision function from `sklearn` [66]. For the halogenase, thiolase, and glycosyltransferase datasets, this is done with leave-one-out cross

validation. For the phosphatase, BKACE, and kinase datasets, to limit the number of trials, we use 10 fold cross validation repeated 5 times. This procedure is repeated for 3 random seeds.

An identical procedure is conducted on the task of substrate discovery, where each enzyme is separately evaluated as its own "task". In this case, the glycosyltransferase, esterase, and phosphatase datasets are evaluated with 5 repetitions of 10 fold cross validation.

### Filtering imbalanced tasks

Certain enzymes have activity on only a few substrates and certain substrates have activity on only a few enzymes. To avoid computing AUPRC values on these tasks, we filter to only incorporate tasks with a maximum fraction of either 0.9 positives or negatives. The remaining tasks can be found in Table B in S1 Text.

### Kinase inhibitor reanalysis

We modify the code from Hie et al. directly to reproduce their GP, GP + MLP models, and add our no-interaction models. GP + MLP models involve first fitting an MLP model followed by a GP to predict residual loss. First, all kinases are converted into features using a pretrained language model [16, 17] and all inhibitors are converted into features using a pretrained JT-VAE [53] or Morgan fingerprints. We create a training set of kinase, inhibitor pairs with labeled $K_d$ values, and establish 3 separate segments of the test data: new kinases (repurposing), new inhibitors (discovery), and new kinase+inhibitor pairs. The data is split into four quadrants and one quadrant is used for training models. GP and linear models are implemented with `scikit-learn` [66] and MLP models are implemented with `Keras` [67], following parameter choices from the original study [42]. Linear regression models are parametrized with $\alpha = 10$ and normalization set to True. Prior to training single-task models, we standardize the regression target values based upon the training set to have a mean of 0 and variance of 1, as we find it helps with stability with less training data.

To test the ability of models to prioritize candidates, we repeat the train/test split and rank the entire test set by predicted $K_d$, using an additional upper confidence bound ($\beta = 1$) metric to adjust rank for the GP + MLP model that uses uncertainty. The top $k = 5$ and $k = 25$ compound-kinase pairs are evaluated by their average true $K_d$. All kinase-inhibitor reanalysis experiments were repeated for five random seeds.

### Pooling strategies

We use the program `Muscle` with default parameters to compute a multiple sequence alignment (MSA) on each dataset for pooling. Because many positions in certain datasets have high coverage, ties are randomly broken when choosing a priority for pooling residue embeddings. This explains the large variance in the glycosyltransferase results shown (Fig 5) taken over several seeds. Coverage and conservation based pooling strategies are sampled for $i \in \{1, 2, 3, 6, 11, 19, 34, 62, 111, 200\}$ pooling residues, each repeated for 3 random seeds.

### Active site pooling

To pool over active sites of proteins, we identify a reference crystal structure within each protein family or super family (Table 1). For each of these crystal structures, we select either an active site bound ligand or active site residue(s) from the literature. We attempt to pool residues within a Cartesian distance from these sites ranging from 3 to 30 angstroms to roughly mirror the number of residues pooled for coverage and conservation methods. Angstrom

shells are calculated using `Biopython` [68] and an in depth description of active sites used can be found in Table A in S1 Text.

## Supporting information

**S1 Text. Fig A: Dataset substrates** 6 exemplar molecule substrates are randomly chosen from each dataset and displayed. **Fig B: Dataset diversity**. Distributions of top-5 enzyme similarity (left) and substrate similarity (right) are shown across enzyme datasets collected. Enzyme similarity is calculated as the percent overlap between two sequences in their respective multiple sequence alignment, excluding positions where both sequences contain gaps. Substrate similarity is computed using Tanimoto similarity between 2048-bit chiral Morgan fingerprints. **Fig C: Enzyme discovery benchmarking with AUCROC**. On the 6 different datasets tested (thiolase datasets used for hyperparameter optimization), K-nearest neighbor baselines with Levenshtein edit distance are compared against feed-forward networks using various featurizations and ridge regression models in terms of AUC ROC performance. ESM-1b features indicate protein features extracted from a masked language model trained on UniRef50 [20]. Concatenation and dot product architectures are indicated with "[{prot repr.}, {sub repr.}]" and "{prot repr.}• {sub repr.}" respectively. Halogenase and glycosyltransferase datasets are evaluated using leave-one-out splits. BKACE, phosphatase, and esterase datasets are evaluated with 5 repeats of 10 different cross validation splits. AUC ROC is calculated using scikit-learn for each substrate task separately before being averaged. Error bars represent the standard error of the mean across 3 random seeds. Each model and featurization is compared to "Ridge: ESM-1b" using a 2-sided Welch T test, with each additional asterisk representing significance at [0.05, 0.01, 0.001, 0.0001] thresholds respectively after applying a Benjamini-Hochberg correction. **Fig D: Full substrate discovery AUC ROC results**. CPI models and single task models are compared on the glycosyltransferase, esterase, and phosphatase datasets, all with 5 trials of 10-fold cross validation. Error bars represent the standard error of the mean across 3 random seeds. Each model and featurization is compared to "Ridge: Morgan" using a 2-sided Welch T test, with each additional asterisk representing significance at [0.05, 0.01, 0.001, 0.0001] thresholds respectively after applying a Benjamini-Hochberg correction. Pretrained substrate featurizations used in "Ridge: JT-VAE" are features extracted from a junction-tree variational auto-encoder (JT-VAE) [53]. Concatenation and dot-product architectures are indicated with "[{prot repr.}, {sub repr.}]" and "{prot repr.}•{sub repr.}" respectively. In the interaction based architectures, "ESM-1b" indicates the use of a masked language model trained on UniRef50 as a protein representation [20]. Models are hyperparameter optimized on a held out halogenase dataset. **Fig E: Substrate Discovery Extended Analysis (i)** Average AUPRC on each individual "enzyme task" is compared between compound protein interaction models and single-task models. Points below 1 indicate substrates on which single-task models better predict enzyme activity than CPI models. CPI models used are "FFN: [ESM-1b, Morgan]" and single-task models are "Ridge: Morgan". **(ii)** AUPRC values from the ridge regression model broken out by each task are plotted against the fraction of active enzymes in the dataset. Best fit lines are drawn through each dataset to serve as a visual guide. **Fig F: MSA and structure based pooling across all datasets tested (i)** Active site, coverage, conservation, and mean pooling are plotted for all 5 enzyme discovery datasets tested. Both AUCROC and AUPRC values are shown. These are compared against the Levenshtein distance baseline (dotted). **(ii)** Equivalent analysis is conducted on the filtered kinase dataset extracted from Davis et al. with MAE, RMSE, and Spearman rank correlation shown [39]. The same hyperparameters are used as set in Fig 2 for ridge regression models. All experiments are repeated for 3 random seeds following the same split evaluation as in other enzyme discovery model benchmarking. **Fig G: Enzyme discovery**

**halogenase prediction results** Ground truth binary enzyme-substrate activities (left) are compared against a single seed of predictions made through cross validation using a single-task ridge regression model (middle) and a CPI based model, FFN: [ESM-1b, Morgan] (right). **Fig H: Enzyme discovery glycosyltransferase prediction results** Ground truth binary enzyme-substrate activities (left) are compared against a single seed of predictions made through cross validation using a single-task ridge regression model (middle) and a CPI based model, FFN: [ESM-1b, Morgan] (right). **Fig I: Enzyme discovery BKACE prediction results** Ground truth binary enzyme-substrate activities (left) are compared against a single seed of predictions made through cross validation using a single-task ridge regression model (middle) and a CPI based model, FFN: [ESM-1b, Morgan] (right). **Fig J: Enzyme discovery esterase prediction results**. Ground truth binary enzyme-substrate activities (left) are compared against a single seed of predictions made through cross validation using a single-task ridge regression model (middle) and a CPI based model, FFN: [ESM-1b, Morgan] (right). **Fig K: Enzyme discovery phosphatase prediction results** Ground truth binary enzyme-substrate activities (left) are compared against a single seed of predictions made through cross validation using a single-task ridge regression model (middle) and a CPI based model, FFN: [ESM-1b, Morgan] (right). **Fig L: Substrate discovery glycosyltransferase prediction results** Ground truth binary enzyme-substrate activities (left) are compared against a single seed of predictions made through cross validation using a single-task ridge regression model (middle) and a CPI based model, FFN: [ESM-1b, Morgan] (right). **Fig M: Substrate discovery esterase prediction results** Ground truth binary enzyme-substrate activities (left) are compared against a single seed of predictions made through cross validation using a single-task ridge regression model (middle) and a CPI based model, FFN: [ESM-1b, Morgan] (right). **Fig N: Substrate discovery phosphatase prediction results** Ground truth binary enzyme-substrate activities (left) are compared against a single seed of predictions made through cross validation using a single-task ridge regression model (middle) and a CPI based model, FFN: [ESM-1b, Morgan] (right). **Table A: Active site structure references used in pooling**. All structure informed pooling strategies require a catalytic center in order to define various angstrom shells of residues to pool over. This table provides the PDB reference crystal structure as well as the reference residues or structural elements used to define the pooling center, from which spherical radii originate. **Table B: Summary of valid substrate and sequence "tasks"**. In each dataset, only certain substrates and sequences are defined as valid "tasks" based upon the balance between active and inactive examples. Each substrate or sequence used for an enzyme or substrate discovery task respectively requires at least 2 positive examples and at a minimum, 10% of examples in that task must be part of the minority class. This table defines the number of valid substrate and sequence tasks. **Table C: Enzyme substrate compatibility models**. Summary and classifications of different models utilized. **Table D: Kinase reanalysis models**. Summary and classifications of different models utilized in our reanalysis of Hie et al. [42]. **Table E: Full enzyme discovery area under the precision recall curve (AUPRC) results**. On the 6 different datasets tested (thiolase datasets used for hyperparameter optimization), K-nearest neighbor baselines with Levenshtein edit distance are compared against feed-forward networks using various featurizations and ridge regression models. Pretrained features ("ESM-1b") indicate protein features extracted from a masked language model trained on UniRef50 [20]. Two compound protein interaction architectures are tested, both concatenation and dot products, indicated with "[{prot repr.}, {sub repr.}]" and "{prot repr.}•{sub repr.}" respectively. Halogenase and glycosyltransferase datasets are evaluated using leave-one-out splits, whereas BKACE, phosphatase, and esterase datasets are evaluated with 5 repeats of 10 different cross validation splits. Average precision is calculated using scikit-learn for each substrate task separately before being averaged. Average values are presented across 3 random seeds ± standard error. **Table F: Full enzyme discovery area under the receiver operating curve (AUC-ROC) results**.

On the 6 different datasets tested (thiolase datasets used for hyperparameter optimization), K-nearest neighbor baselines with Levenshtein edit distance are compared against feed-forward networks using various featurizations and ridge regression models. ESM-1b features indicate protein features extracted from a masked language model trained on UniRef50 [20]. Two compound protein interaction architectures are tested, both concatenation and dot products, indicated with "[{prot repr.}, {sub repr.}]" and "{prot repr.}•{sub repr.}" respectively. Halogenase and glycosyltransferase datasets are evaluated using leave-one-out splits, whereas BKACE, phosphatase, and esterase datasets are evaluated with 5 repeats of 10 different cross validation splits. AUC ROC is calculated using scikit-learn for each substrate task separately before being averaged. Average values are presented across 3 random seeds ± standard error. **Table G: Full substrate discovery area under the precision recall curve (AUPRC) results**. CPI models and single task models are compared on the glycosyltransferase, esterase, and phosphatase datasets, all with 5 trials of 10-fold cross validation. Each model and featurization is compared to "Ridge: Morgan" using a 2-sided Welch T test, with each additional asterisk representing significance at [0.05, 0.01, 0.001, 0.0001] thresholds respectively after applying a Benjamini-Hochberg correction. Pretrained substrate featurizations used in "Ridge: JT-VAE" are features extracted from a junction-tree variational auto-encoder (JT-VAE) [53]. Two compound protein interaction architectures are tested, both concatenation and dot-product, indicated with "[{prot repr.}, {sub repr.}]" and "{prot repr.}•{sub repr.}" respectively. In the interaction based architectures, ESM-1b indicates the use of a masked language model trained on UniRef50 as a protein representation [20]. Average precision is calculated using scikit-learn for each substrate task separately before being averaged. Models are hyperparameter optimized on a held out halogenase dataset. Values represent mean values across 3 random seeds ± standard error. **Table H: Full substrate discovery area under the receiver operating curve (AUC-ROC) results**. CPI models and single task models are compared on the glycosyltransferase, esterase, and phosphatase datasets, all with 5 trials of 10-fold cross validation. Each model and featurization is compared to "Ridge: Morgan" using a 2-sided Welch T test, with each additional asterisk representing significance at [0.05, 0.01, 0.001, 0.0001] thresholds respectively after applying a Benjamini-Hochberg correction. Pretrained substrate featurizations used in "Ridge: JT-VAE" are features extracted from a junction-tree variational auto-encoder (JT-VAE) [53]. Two compound protein interaction architectures are tested, both concatenation and dot-product, indicated with "[{prot repr.}, {sub repr.}]" and "{prot repr.}•{sub repr.}" respectively. In the interaction based architectures, "ESM-1b" indicates the use of a masked language model trained on UniRef50 as a protein representation [20]. Models are hyperparameter optimized on a held out halogenase dataset. Values represent mean values across 3 random seeds ± standard error.
(PDF)

## Acknowledgments

The authors thank Brian Hie, Bryan Bryson, and Bonnie Berger for discussion around kinase inhibitor screening. The authors thank Karine Bastard, Marcel Salanoubat, Ben Davis, and Charlie Fehl for helpful discussions around their respective experimental screens. The authors acknowledge the MIT SuperCloud and Lincoln Laboratory Supercomputing Center for providing HPC resources.

## Author Contributions

**Conceptualization:** Samuel Goldman, Connor W. Coley.

**Data curation:** Samuel Goldman.

**Funding acquisition:** Connor W. Coley.

**Investigation:** Samuel Goldman, Ria Das.

**Methodology:** Samuel Goldman, Kevin K. Yang, Connor W. Coley.

**Project administration:** Connor W. Coley.

**Software:** Samuel Goldman, Ria Das.

**Supervision:** Connor W. Coley.

**Visualization:** Samuel Goldman.

**Writing – original draft:** Samuel Goldman, Connor W. Coley.

**Writing – review & editing:** Samuel Goldman, Kevin K. Yang, Connor W. Coley.

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
