## [Decision Letter · Decision Letter 0]

12 Dec 2021

Dear Prof. Coley,

Thank you very much for submitting your manuscript "Machine learning modeling of family wide enzyme-substrate specificity screens" for consideration at PLOS Computational Biology. As with all papers reviewed by the journal, your manuscript was reviewed by members of the editorial board and by several independent reviewers. The reviewers appreciated the attention to an important topic. Based on the reviews, we are likely to accept this manuscript for publication, providing that you modify the manuscript according to the review recommendations.

Sincerely,

Rachel Kolodny

Associate Editor

PLOS Computational Biology

Nir Ben-Tal

Deputy Editor

PLOS Computational Biology

[LINK]

Reviewer's Responses to Questions

**Comments to the Authors:**

Reviewer #1: Goldman et al carry out a thorough evaluation of how machine learning might be used to predict the substrate specificity of enzymes for biocatalysis. They present two important and relevant questions; can substrate specificity be predicted from the known substrate scope of an enzyme, and can enzymes active against a given substrate be predicted from data on other enzymes. An interesting finding in this work is that utilising the data for multiple enzymes against multiple substrates does not outperform training on either a single substrate (with multiple enzymes), or a single enzyme (against multiple substrates). It might be expected that utilising all the data would result in better predictions, but this is shown not to be the case. This sets the scene for future work in this area. The authors also demonstrate an interesting approach for creating a representation of only the active site of an enzyme, which in some cases shows improved predictive performance.

I found the paper to be well written and structured in a logical way, and I enjoyed reading it. The authors make a valuable contribution to the field, and I believe the work will be of interest to the readership of PLOS Computational Biology. I have made a few comments which the authors might like to consider, but I otherwise recommend this work for publication.

Comments

It might be helpful to briefly note in the methods section that the thresholds for binarization are presented in the supporting information.

Why did the authors choose ESM-1b (other than being ‘state of the art’). Is it expected that ESM-1b will perform better than other representations such as UniRep or SeqVec? Were other representations investigated? Would there be any benefit to including the choice of sequence representation as a hyper-parameter in a machine learning pipeline? A brief description of why ESM-1b was chosen would be useful.

For the CPI models, what is the rational for taking the dot product of the substrate and sequence representations for predictions? Also, how were the representations projected into smaller and equal length vectors (line 148)?

Why was Levenshtein distance used for the KNN studies? Could a standard percentage sequence identity as is commonly reported in BLAST searches be used instead, and might this perform better given it takes into account similarities between different amino acids via a BLOSUM scoring matrix? For sequence similarity networks often an ‘alignment score’ is used, which may also perform better? (https://doi.org/10.1016/j.bbapap.2015.04.015)

Line 187 to 194 – should Fig 2E be referenced here?

SI Fig 7 onwards – what do the colours mean?

Minor typos

Line 292 – Typo in word most

Line 345 – cross leave one out cross validation

Reviewer #2: The review is uploaded as an attachment.

Reviewer #3: General Comments:

In this work, the authors use protein language model embeddings, along with substrate encodings to predict compound protein interactions. They work with multiple top models which incorporate multiple chemical encoding methods and concatenation approaches and tested with different dataset. The use of protein language model, chemical encodings and different machine learning models shows an innovative integration of many works in the literature. The proposed structure-based strategy for pooling residue representations improves CPI predictions, which has many applications in biocatalysis. This work is very well written and will form an extremely important contribution to not only the ML community but also for the biocatalysis, protein engineering and functional genomics community. I should also commend, applaud and thank the authors for providing the datasets that were used in the paper. That is significant contribution though the code for some of the methods were not clear in the GitHub repository. Please see if you can add these details as well. Taken together this paper is highly valuable though it can be strengthened further based on the following comments.

Major Comments:

The introduction provides a nice description of the background but it could use more information on the protein sequence modeling such as the NLP methods used for prediction. I would urge authors to cut down the background on the enzyme side and add some paragraphs describing the protein sequence modeling (the different language models used and their success in enabling protein property prediction) which forms the foundation for their use in this specific sequence to function task. Similarly, there is some background on the substrate representations that could be relevant. This could also be a place to rationalize their choice of language model (ESM1b) and the substrate representations (JT-VAE).

The dataset published on GitHub is clear and well-organized. And results of CPI as well as structure-based classification models are clearly shown. However, results of regression models are not fully shown. For example, in figure 5B, Spearman’s R is only plotted for Kinase dataset, while phosphatase dataset also contains activities (numeric values) for regression models. The Spearman’s R’s for predicting CPI of phosphatases and halogenases activities are not shown or discussed.

166-168: Why are the hyperparameters tuned based on a small dataset (thiolase) rather than performing hyperparameter tuning on the largest dataset (i.e., phosphatases)? And what is the predictive performance on thiolase dataset after hyperparameter tuning?

174-178: Are there explanations on why CPI models do not outperform single-task models given that CPI models have been trained on both enzymes and substates as inputs and a greater amount of data? What is the AUPRC of CPI models using KRR as top model instead of FFN? Is this due to dissimilarities in the substrate structures or enzyme sequences maybe?

198-200: Similarly, what is the predictive performance on halogenase dataset after hyperparameter tuning?

Figure 5B: It seems that those AUPRC’s shown are for the sequence discovery tasks, according to baseline values in the AUPRC plots. For the last figure, the 0.42 Spearman’s R baseline for kinase dataset seems to be that of the substrate discovery task. Please specify this in the caption. Also, kinase’s sequence discovery task results are also interesting and should be discussed.

Minor Comments:

58-60: Consider citing this reference for enzyme identification:

https://link.springer.com/article/10.1186/1752-0509-4-35

127-128: The argument that the novel substrates and sequence prediction is tough is well taken but nevertheless the authors could add some data to show this.

189-190: Can the authors please refer to this plot 2E here ? May be rewrite “Plotting the AUPRC metric as a function of number of actives”

259-260: Authors point to this important issue in the field. There has been some recent work that has indicated that random emeddings can also lead to useful predictions sometimes mitigating the value of language models and perhaps authors can cite this work here.

269-270: Could you provide more details about how the different “reference structures" you provided are combined for identifying active site in your proposed methods of structure-based active site pooling strategy?

**Have the authors made all data and (if applicable) computational code underlying the findings in their manuscript fully available?**

Reviewer #1: Yes

Reviewer #2: Yes

Reviewer #3: **No: **The Github page does not have the details of some parts of the code needed to reproduce the results though it has the datasets information.

PLOS authors have the option to publish the peer review history of their article (what does this mean?). If published, this will include your full peer review and any attached files.

Reviewer #1: No

Reviewer #2: No

Reviewer #3: No

Figure Files:

Data Requirements:

Reproducibility:

References:

---

## [Editor Report · Decision Letter 1]

21 Jan 2022

Dear Prof. Coley,

We are pleased to inform you that your manuscript 'Machine learning modeling of family wide enzyme-substrate specificity screens' has been provisionally accepted for publication in PLOS Computational Biology.

Best regards,

Rachel Kolodny

Associate Editor

PLOS Computational Biology

Nir Ben-Tal

Deputy Editor

PLOS Computational Biology

---

## [Editor Report · Acceptance letter]

7 Feb 2022

PCOMPBIOL-D-21-01963R1 

Machine learning modeling of family wide enzyme-substrate specificity screens

Dear Dr Coley,

I am pleased to inform you that your manuscript has been formally accepted for publication in PLOS Computational Biology. Your manuscript is now with our production department and you will be notified of the publication date in due course.

With kind regards,

Zsofia Freund
